# Effect of Random Nanostructured Metallic Environments on Spontaneous Emission of HITC Dye

**DOI:** 10.3390/nano10112135

**Published:** 2020-10-27

**Authors:** Sangeeta Rout, Zhen Qi, Ludvig S. Petrosyan, Tigran V. Shahbazyan, Monika M. Biener, Carl E. Bonner, Mikhail A. Noginov

**Affiliations:** 1Center for Materials Research, Norfolk State University, Norfolk, VA 23504, USA; s.rout@spartans.nsu.edu (S.R.); cebonner@nsu.edu (C.E.B.); 2Lawrence Livermore National Laboratory, Livermore, CA 94550, USA; qi2@llnl.gov (Z.Q.); biener3@llnl.gov (M.M.B.); 3Department of Physics, Jackson State University, Jackson, MS 39217, USA; lyudvig1977@gmail.com (L.S.P.); shahbazyan@jsums.edu (T.V.S.)

**Keywords:** Au nanofoams, concentration quenching, emission kinetics, energy transfer

## Abstract

We have studied emission kinetics of HITC laser dye on top of glass, smooth Au films, and randomly structured porous Au nanofoams. The observed concentration quenching of luminescence of highly concentrated dye on top of glass (energy transfer to acceptors) and the inhibition of the concentration quenching in vicinity of smooth Au films were in accord with our recent findings. Intriguingly, the emission kinetics recorded in different local spots of the Au nanofoam samples had a spread of the decay rates, which was large at low dye concentrations and became narrower with increase of the dye concentration. We infer that in different subvolumes of Au nanofoams, HITC molecules are coupled to the nanofoams weaker or stronger. The inhibition of the concentration quenching in Au nanofoams was stronger than on top of smooth Au films. This was true for all weakly and strongly coupled subvolumes contributing to the spread of the emission kinetics. The experimental observations were explained using theoretical model accounting for change in the Förster radius caused by the strong energy transfer to metal.

## 1. Introduction

The most recent two decades witnessed a rapid development of the interrelated research fields of plasmonics [1,2], nanophotonics [3,4], metamaterials [5,6,7], and strong coupling of light and matter at the nanoscale [8,9]. While originally designed to manipulate incident and emitted light [10,11,12], photonic metamaterials were recently shown to control scores of physical phenomena, including Förster energy transfer [13,14,15,16], van der Waals interactions [17,18], surface potentials [19], and chemical reactions [20,21,22,23,24,25], which lie outside the traditional electrodynamics domain. The Förster energy transfer (including concentration quenching of spontaneous emission), which is of particular interest to this study, has been researched in vicinity of mirrors [26,27,28] and plasmonic nanoparticles [29], in Fabry-Perot cavities [13], and on top of metamaterials [15]. Although debates of the mechanisms of the energy transfer in the metal/dielectric environments above did not settle yet, even less is known about the energy transfer in random, scattering, and multi-phase matrixes. To our knowledge, this is the first experimental study of the energy transfer and concentration quenching in random nanostructured metal/dielectric materials.

## 2. Experimental Samples

The metallic and dielectric host matrixes and substrates in our study included (i) nanostructured Au nanofoams (fabricated as discussed in Methods), Figure 1, (ii) Au films deposited on glass (see Methods), and (iii) glass (control samples). The HITC:PMMA dye-doped polymers, spin coated onto the substrates above, were 80 ± 10 nm thick and had the concentrations of HITC (2-[7-(1,3-dihydro-1,3,3-trimethyl-2H-indol-2-ylidene)-1,3,5-heptatrienyl]-1,3,3-trimethyl-3H-indoliumiodide) dye ranging between 3.2 g/L and 36.5 g/L in solid state. (1 g/L of HITC in poly (methyl methacrylate) PMMA is equivalent to 0.00186 mol/L and 1.23 × 10^18^ cm^−3^). The fabrication of the dye-doped polymeric thin films is discussed in Methods.

## 3. Spectroscopic Properties

As it was shown in a recent study [16] that the absorption and excitation bands of HITC:PMMA films on glass have the maxima at 762 nm, and the emission band (the mirror image of the absorption band) has its maximum at 772 nm, Figure 2. The emission bands in HITC:PMMA films deposited on top of Au had a slight blue shift, which was much smaller than the emission bandwidth. (This phenomenon is a subject of a separate study to be published elsewhere.) The absorption spectra were taken using the spectrophotometer Lambda 900 (from PerkinElmer, Waltham, MA, USA) and the emission and excitation spectra were taken using the spectrofluorimeter (Fluorolog 3 from Horbia, Kyoto, Japan). Similar results were obtained in the present study, in which the same instruments were used in absorption, reflection and emission measurements.

## 4. Emission Kinetics Measurements

In the emission kinetics measurements, the samples were excited at λ = 795 nm with ~150 fs pulses of the mode locked Ti:sapphire laser, Mira 900 (from Coherent, CA, USA). The diameter of the laser spot on the sample was ~2.5 mm and the average power (at 76 MHz repetition rate) was ~60 mW. The emission kinetics were recorded using the visible and near-infrared Streak Camera (Model C5680 from Hamamatsu, Japan). The measured width of the laser pulse, determined by the jitter of the laser and wide-open entrance slit of the streak camera was ~100 ps. A combination of the interference and long pass filters was used to block the laser light and transmit the HITC emission at λ ≥ 850 nm. Please note that some short-pulsed light (presumably laser light) leaked through the filters if the samples were scattering, e.g., Au nanofoams with or without dye, but not if smooth Au or glass substrates were used. To make sure that the scattered light did not interfere with our studies of the HITC emission, the kinetics were analyzed starting from 150 ps after the maximum of the laser pulse (after the scattered light was over).

## 5. Emission Kinetics in HITC:PMMA Films Deposited on Glass

The HITC emission kinetics in the samples with low dye concentrations (*n* ≤ 3.2 g/L), deposited on glass were nearly single exponential (Figure 3a). However, they noticeably shortened and deviated from exponential functions at larger values of *n*, see Figure 3b (*n* = 36.5 g/L). This is the characteristic signature of a concentration quenching (energy transfer to acceptors) [16]. The effective emission decay rates, obtained by fitting experimental emission kinetics with single exponential functions, can be adequately fitted with the formula (*A* + *W*) + *γn*^2^ (Figure 4), where *A* and *W* are the rates of intra-central radiative and non-radiative decay and *γn*^2^ is the energy transfer rate (see fitted values *A* + *W* and *γn*^2^ in Figure 5). The latter quadratic polynomial behavior, in agreement with Ref. [16], is consistent with two possible scenarios: (i) quenching centers involve pairs of aggregated dye molecules and the energy transfer to quenching centers is direct or (ii) the concentration of quenching centers is proportional to the concentration of dye molecules, but the energy transfer is migration-assisted [30].

## 6. Emission Kinetics in HITC:PMMA Films Deposited on Smooth Au Films

As with HITC:PMMA films deposited on glass, the HITC emission kinetics measured on top of smooth thermally deposited Au films were relatively long and nearly single exponential at small dye concentrations (*n* = 3.2 g/L), Figure 3a. However, they shortened and deviated from exponential functions at large values of *n* (*n* = 36.5 g/L), Figure 3b. The latter shortening was not as strong as that on top of glass substrates, manifesting inhibition of the concentration quenching in vicinity of gold [16]. The extracted value *A + W* was nearly the same as on top of glass, while the value *γn*^2^ was nearly two times smaller, Figure 5. This is the first important result of this study.

The ratios of the decay rates in HITC:PMMA films on top of Au and on top of glass are depicted in Figure 6. One can see that at large dye concentrations, *n* ≥ 20 g/L, the ratio is smaller than unity, due to inhibition of the concentration quenching. On the other hand, at low dye concentrations (*n* ≤ 20 g/L), when the concentration quenching is modest or small, the emission kinetics of HITC on smooth Au films, expectedly, become faster than those on glass (due to energy transfer to metal) [31]. In agreement with the Theoretical Modeling section (below), at the film thickness equal to 80 nm, this effect is modest, ~15%, comparable to the error bar in this particular experiment, Figure 6. Therefore, there is no contradiction, within the experimental error bar, between the results of Figure 5 and Figure 6.

In another particular experiment, we measured emission intensities *I_0_* right after the pumping pulse, before any noticeable decay of the excited state concentration could take place. (In this particular measurement, we made sure that no parasitic scattered laser light got mixed with the emission signal.) These initial intensities are proportional to the product of the Einstein coefficient *A* and the concentration of excited molecules *n**. (The latter was assumed to be proportional to the fraction of the pumping absorbed.) The scatter of the data points was large, partly because the samples with different dye concentrations were measured in different days and the precise optical alignment was not preserved from measurement to measurement. To reduce the data scatter, the initial emission intensities *I_0_* measured on top of Au films were divided by those on top of glass (taken in the same day), Figure 7. The increase of the latter ratios with increase of the dye concentration is in qualitative agreement with the reduction of the concentration quenching in vicinity of metal. However, although the emission decay rates and the initial emission intensities are related quantities, they are not directly proportional to each other (the former depends on the non-radiative decay rate and the latter does not).

The demonstrated inhibition of the concentration quenching in vicinity to gold is qualitatively similar to that in HITC:PMMA films on top of silver reported in Ref. [16]. This phenomenon is highly intriguing, since the common perception is that the molecular emission is quenched in vicinity of a metal, because of the energy transfer to a metal [32]. On the contrary, the experimentally observed behavior is opposite: *reduction* of the emission kinetics rate in vicinity of metal. This seeming controversy will be explained in the Theoretical Modeling section.

## 7. Emission of HITC:PMMA Films Spin Coated onto Au Nanofoam Samples

In the series of experiments discussed in this section, HITC:PMMA/DCM solutions were spin coated onto Au nanofoams described above (DCM, dichloromethane, is the solvent). In these samples, the dye, whose spatial distribution was difficult to quantify, predominantly penetrated into the nanofoam’s volume. The presence of HITC dye was barely seen at *λ*~770 nm in the reflection spectra of dye-impregnated nanofoams, *n* = 36.5 g/L (Figure 8, trace 2), as compared to nanofoams without dye (Figure 8, trace 1). At the same time, the presence of dye was clearly seen in the reflection spectrum of the HITC:PMMA film (*n* = 36.5 g/L, 80 nm) deposited on smooth Au film (Figure 8, trace 4), as compared to the reflection spectrum of a pristine Au film without dye (Figure 8, trace 3). (Similar observations have been made at multiple dye concentrations.)

The emission decay kinetics in the dye-doped Au nanofoam samples were *highly inhomogeneous.* This is one of the most important results of this study. Thus, emission kinetics measured in different locations on the sample could have strongly different rates. The spread of the emission kinetics in the 12.7 g/L sample is depicted in Figure 3c. The black traces in Figure 3c approximately correspond to the upper (slow decay) and the lower (fast decay) boundaries of the emission kinetics spread. The characteristic lateral scale of the emission non-uniformity (the effective sizes of domains with more or less homogeneous kinetics) was ~1 mm. Such spatial non-uniformity was never observed on top of glass or smooth Au films.

At small dye *concentration, n = 3.2 g/L*, the HITC:PMMA emission kinetics on top of glass and smooth Au films are almost identical (Figure 3a), the concentration quenching is weak [16], and the effect of a smooth gold substrate on the dye emission is relatively small (Figure 6). In the Au nanofoam sample, the kinetics corresponding to the upper boundary of the emission kinetics spread are the same as those on top of smooth gold and glass. At the same time, the kinetics corresponding to the lower boundary of the spread have significantly shorter (nearly threefold) decay times. We infer that in different subvolumes of the Au nanofoam, HITC molecules are coupled to the nanofoam weaker or stronger. Correspondingly, unquenched emission kinetics are due to HITC molecules modestly or weakly coupled to the Au nanofoam, while the shortened emission kinetics are due to the dye molecules, which are strongly coupled to and quenched by the Au nanofoam. (Here by strong coupling we do not mean the light-matter interaction resulting in the Rabi splitting [8]).

To determine whether increase of the decay rates of the dye molecules coupled to Au nanofoams was due to enhancement of spontaneous emission *A* or non-radiative decay *W*, we analyzed multiple emission kinetics corresponding to the kinetics spread (similar to those in Figure 3c) and plotted the initial emission intensities *I* (~A) against the emission rates *A + W*, Figure 9. (At *n* = 3.2 g/L, the concentration quenching *γn*^2^ was negligibly small.) The reduction of *A* (~*I*) with increase of *A + W* suggests that the increase of the decay-time is due to increase of *W* rather than *A* (or, if both *A* and *W* are increased, the increase of *W* is larger than the increase of *A*). This behavior is consistent with the theoretical prediction [31].

At intermediate dye concentration, *n* = 12.7 g/L, the emission kinetics on glass and on smooth Au films are close to each other (Figure 3d), although, they are shorter than those in the *n* = 3.2 g/L sample (Figure 3a). This signifies modest concentration quenching, whose inhibition is balanced by the energy transfer to metal. The latter emission kinetics are close to the upper (slower) boundary of the kinetics spread in Au nanofoams (which is narrower than that at low dye concentration) and much above the lower (fast) boundary of the kinetics spread, Figure 3d.

At high dye concentration, *n* = 36.5 g/L, the concentration quenching on top of glass and its inhibition in vicinity of a smooth Au film are strong. Correspondingly, the emission kinetics on top of smooth gold are significantly different from those on glass, Figure 3b. In Au nanofoams, the spread of the emission kinetics becomes small and both upper and lower boundaries of the spread lie above the emission kinetics on glass and smooth gold, see Figure 3b and Figure 4. The latter observation suggests that the inhibition of the concentration quenching in Au nanofoams (in both strongly and modestly coupled subvolumes) is stronger than that on top of smooth Au films. This important result of our study is discussed in detail below.

The decay rates corresponding to the upper (slow decay) boundary of the emission kinetics spread and their fitting with the formula (*A* + *W*) + *γn*^2^ are depicted in Figure 4 (blue triangles). The corresponding fitted values *A + W* and *γn*^2^ are shown in Figure 5. The determined value *A + W* is close to those on top of glass and smooth Au films. Correspondingly, the emission kinetics on glass, smooth Au film and Au nanofoam (slow decay boundary of the kinetics spread) are nearly the same at small dye concentrations, when the concentration quenching is negligibly small, Figure 3a. The determined value *γn*^2^ is nearly twice smaller than that in smooth Au samples. Therefore, at high dye concentrations, the emission kinetics in Au nanofoams are slower than those in smooth Au samples, Figure 3b. This proves that Au nanofoams inhibit concentration quenching much stronger than smooth Au films do, as discussed above. This experimental result can be explained by large surface area of Au nanofoams and relatively small molecule-to-metal distances in nanofoam samples (see the Theoretical Modeling section for details).

The decay rates corresponding to the low (fast decay) boundary of the emission kinetics spread have a strong concentration-independent contribution, presumably originating from the strong coupling of dye molecules with metal and the energy transfer to metal. The fitting of the decay rates with the formula *(A + W)* + *γn*^2^ and the extracted values *(A + W)* and *γn*^2^ are depicted in Figure 4 and Figure 5. Noteworthy is the very large value *A + W*, which is four times larger than the analogous values in the other samples studied, Figure 5. This decay mechanism is so strong that it inhibits the concentration quenching almost completely (see Theoretical Modeling). Therefore, the slope of the corresponding trace in Figure 4 is very small, in agreement with the Theoretical Modeling, and the decay rate at high concentration is nearly the same as that at low concentration. As the decay rates corresponding to the upper (slow decay) boundary of the emission kinetics spread increase with the growth of the dye concentration and the decay rates corresponding to the lower (fast decay) boundary of the emission kinetics spread practically do not, the spread of the emission kinetics is getting smaller with increased concentration. This is another important result of this study.

## 8. Theoretical Modeling

Here we outline the theoretical model that we used to interpret the experimental data. The inhibition of the concentration quenching in vicinity of metallic surfaces at high dye concentrations has been explained in Ref. [31] by a reduction of the Förster radius $R_{F}$ that controls the rate of the Förster resonant energy transfer (FRET) from donors to acceptors. Specifically, near a metallic structure, the Förster radius is modified as $R_{F}^{6}/R_{0}^{6}=\gamma_{sp}/\gamma_{d}$, [33] where $R_{0}$ is the Förster radius in the absence of metal, $\gamma_{sp}$ is isolated donor’s spontaneous decay rate and $\gamma_{d}$ is the donor’s full decay rate that includes energy transfer to the metal [34]. Therefore, for a given donor-acceptor separation R, the normalized FRET rate $\gamma_{FRET}/\gamma_{sp}=R_{F}^{6}/R^{6}$ is reduced as well. Correspondingly, the emission kinetics for a donor in the presence of randomly distributed acceptors with concentration $n_{a}$ is described by $I\left(t\right)∼e^{-\gamma_{d}t-n_{a}V_{F}\sqrt{\pi \gamma_{sp}t}}$, where $V_{F}=4\pi R_{F}^{3}/3$ is the Förster volume [31]. Please note that as the donor distance to metal decreases, the first term in the exponent, characterizing the quenching by metal, rapidly increases following the characteristic $\gamma_{d}\propto d^{-3}$ dependence [34]. At the same time, the second term in the exponent, responsible for the concentration quenching, decreases near the metal following the dependence $V_{F}\propto \sqrt{\gamma_{sp}/\gamma_{d}}$ due to a reduction of the Förster radius. As a result, near the metallic surface, the quenching by metal is enhanced whereas the concentration quenching is reduced. For high acceptor concentrations, the latter effect can be relatively strong, resulting in an effective reduction of the overall decay rate [31].

In Figure 10, we present the results of numerical calculations, performed using the above theoretical approach, for concentration dependence of the effective decay rate $\gamma_{eff}$, which is obtained by representing the emission kinetics for HITC:PMMA film of thickness L on top of Au surface as a single exponential decay (we adopted the Förster radius $R_{0}\approx 5$ nm and the wavelength 800 nm). For convenience, we adopt the parameter $N_{a}=n_{a}V_{0}$ that characterizes the number of acceptors within the Förster volume (for $R_{0}=5$ nm, $N_{a}=1$ corresponds to $n_{a}\approx 1.7$ g/L). Please note that the acceptor concentration is expected to be substantially smaller than the donor concentration. For HITC:PMMA film on glass, the calculated effective rate increases super-linearly with concentration, starting with the value $\approx \gamma_{sp}$ at a low concentration ($N_{a}=0.1$) and reaching the value $\gamma_{eff}\approx 60\gamma_{sp}$ at $N_{a}=6$, indicating strong concentration quenching. These calculations carried within our model of Förster radius suppression near metallic surface [34], are consistent with the experimental data (compare Figure 4 and Figure 10).

When HITC:PMMA film of thickness L is deposited on top of Au substrate, the calculated decay rate still increases with the dye concentration, but at high concentration its value is smaller than that for film on top of glass substrate, which is consistent with the inhibition of the concentration quenching observed experimentally (see Figure 4 and Figure 10). At small dye concentrations, when the quenching by metal is dominant, the decay rate on top of metal is larger than on top of glass, due to higher average rate of the energy transfer to metal. However, at high dye concentrations, when the concentration quenching is dominant, the overall calculated decay rate is reduced in thinner films due to a stronger suppression of FRET in close vicinity to the metal. With increasing dye concentration, the ratio of calculated decay rates on top of metal and of glass, calculated at the same dye concentrations, steadily decreases, as illustrated in Figure 10b (consistent with the experiment, Figure 6), and the critical concentration, at which this ratio becomes <1, depends on the film thickness L that determines the average energy transfer rate to the metal.

Let us now link our model calculations performed for HITC:PMMA dyes on top of flat metal surfaces to the emission kinetics observed in Au nanofoams. We note that a significant spread between fast-decaying and slow-decaying emission kinetics in Figure 3 and between the corresponding decay rates in Figure 4 measured in different local subvolumes of the Au nanofoam samples, can be attributed to strong variations of donor-to-metal energy transfer rates. While it is not feasible to model quantitatively such variations of $\gamma_{d}$, the effective decay rates $\gamma_{eff}$ in different subvolumes can be qualitatively modeled by a HITC:PMMA film with changing, in different subvolumes, thickness L on top of a flat Au surface to account for the change of γ_d_. This model is supported by a similarity between Figure 10a, which shows the effective decay rates calculated for different film thicknesses L, and Figure 4, where the observed fastest and slowest decay rates in nanofoams can be associated with those for L=10 nm and L=40 nm films on top of flat Au surface. Although the variations of decay rates for nanofoams are likely stronger, especially for low dye concentrations, the main trends observed in Figure 4 are qualitatively reproduced within our model. We, therefore, can conclude that slowing down of the emission kinetics in nanofoams, albeit stronger than on flat metal surfaces, is driven by the same mechanism of FRET suppression in the vicinity of metal.

## 9. Summary

To summarize, in this work, we studied emission kinetics of HITC laser dye, embedded in the PMMA polymeric matrix, on top of smooth Au films and in random nanostructured metal-dielectric environments (Au nanofoams). The HITC:PMMA films deposited onto glass substrates served as control samples.

The concentration quenching of luminescence of highly concentrated dye on top of glass substrates, in accord with Ref. [16], is explained in terms of the energy transfer to acceptors formed by (i) pairs of dye molecules (at direct donor-acceptor energy transfer) or (ii) single dye molecules interacting with the host polymeric matrix (at migration-assisted energy transfer [30]).

The concentration quenching is inhibited on top of smooth Au films, in agreement with Ref. [16], where similar effects were observed on top of Ag films and Ag-based lamellar metal-dielectric metamaterials.

The emission kinetics recorded in different local spots of the Au nanofoam samples had very large spreads, which was never observed on top of smooth gold substrates. We infer that in different subvolumes of the Au nanofoam, HITC molecules are coupled to the nanofoam weaker or stronger.

The spread of the emission kinetics was particularly large at small dye concentrations *n*. It became smaller with increase of *n*, as the result of the interplay between the energy transfer to metal and the concentration quenching.

The inhibition of the concentration quenching was stronger in Au foams (in both strongly coupled subvolumes and weakly coupled subvolumes) than on top of smooth Au films. The most likely explanations of this difference include (i) large surface area and (ii) small molecule-to-metal distances in Au nanofoam samples.

The inhibition of concentration quenching in smooth Au film and Au nanofoam samples was explained using a theoretical model that is based on reduction of the Förster radius in the presence of strong energy transfer to the metal [34]. While this model has been used previously to interpret a similar concentration quenching suppression observed for HITC dyes on top of flat Ag film [31], here it is adopted for Au nanofoams exhibiting varying emission kinetics originating from different subvolumes. Specifically, each subvolume was represented by a flat surface covered by the HITC film of some effective thickness that changed across the sample. The results numerical calculations of the corresponding effective decay rates have been found to be in a good qualitative agreement with the experimental data.

In this study, we attributed most of the experimental observations to metal/dielectric environments influencing the dye molecules. The effects of purely dielectric environments on the emission kinetics of HITC dye is the subject of the further study to be published elsewhere.

## 10. Methods

### 10.1. Fabrication of Au Nanofoams

Disk-shaped nanoporous gold samples (Au nanofoams) of diameter ~6 mm and thickness ~200 μm, were prepared by selective dealloying of Ag_70_Au_30_ alloy disks in 68% HNO_3_ solution for 2 days at room temperature, as described in detail in the literature [35,36]. This treatment resulted in Au nano-foams, with characteristic sizes of ligaments and voids of approximately 50–100 nm as shown in Figure 1. To prevent charge transfer between Au and HITC dye, the Au nanofoam samples were coated with Al_2_O_3_ films using the well-established trimethylaluminum and H_2_O (AlMe_3_/H_2_O) atomic layer deposition (ALD) process in a warm wall reactor at a wall and stage temperature of 125 °C, as described previously [37,38,39]. Long exposure and purge times of 300 s each were used to ensure uniform coatings throughout the porous material. The resulting growth rate for Al_2_O_3_ films was 0.25 nm per cycle. The thickness of the Al_2_O_3_ coating was 8–10 nm.

### 10.2. Thermal Deposition of Au Films

Glass substrates, 22 mm × 22 mm × 0.17 mm, were acquired from VWR International. Gold films were deposited onto glass substrates using the thermal vapor deposition apparatus (Nano 36 from Kurt J. Lesker, USA). At the characteristic thickness of ~80 nm and roughness ±5 nm [40], Au films were nearly ideal reflectors in the spectral range of interest, λ > 700 nm.

### 10.3. Dye-Doped Polymeric Films

In preparation of the dye-doped polymeric films, 2-[7-(1,3-dihydro-1,3,3-trimethyl-2H-indol-2-ylidene)-1,3,5-heptatrienyl]-1,3,3-trimethyl-3H-indoliumiodide (HITC) dye and poly (methyl methacrylate) (PMMA) polymer were dissolved in dichloromethane (DCM) and sonicated for t = 90 min at T = 22 °C. The dye concentrations (in the solid state) varied between 3.2 g/L and 36.5 g/L. (1 g/L of HITC in PMMA is equivalent to 0.00186 mol/L or 1.2267 × 10^18^ cm^−3^). The HITC:PMMA doped solutions were spin coated onto the glass, Au films, and Au nanofoam substrates using the Spin-Coater (Model P6700 from Specialty Coating Systems, USA). The thickness of all polymeric and metallic films was measured using the profilometer Dektak XT from Bruker, USA. The characteristic thickness of the HITC:PMMA films on smooth substrates was 80 ± 10 nm.

## Figures and Tables

**Figure 1 nanomaterials-10-02135-f001:**
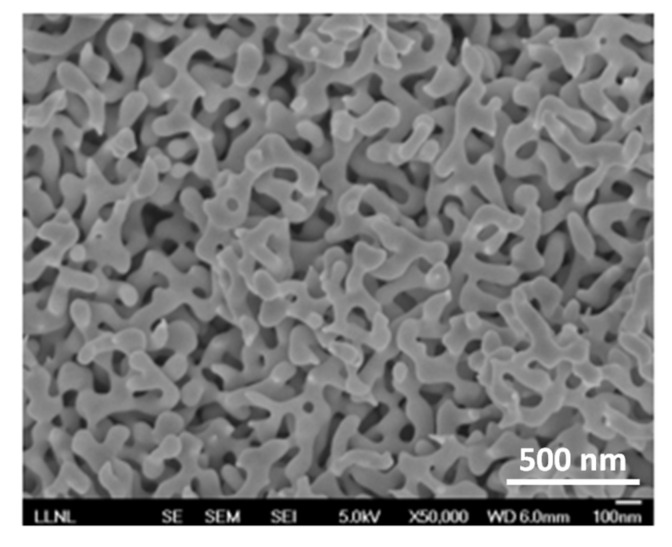
Scanning Electron Microscope (SEM) image of the Au nanofoam.

**Figure 2 nanomaterials-10-02135-f002:**
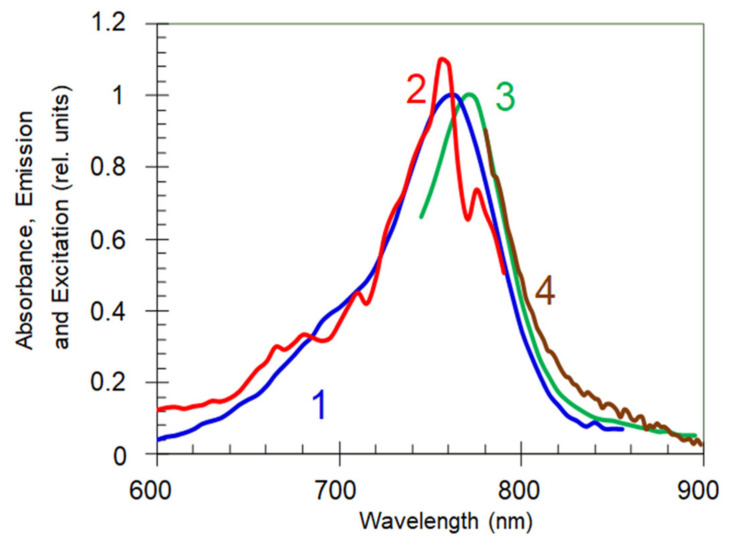
Absorption (trace 1), excitation (trace 2) and cw emission (trace 3) spectra of HITC:PMMA film on the glass substrate at the dye concentration equal to 8.5 g/L. Trace 4: emission spectrum at short-pulse pumping. Reproduced with permission of [16] 2019. The Optical Society.

**Figure 3 nanomaterials-10-02135-f003:**
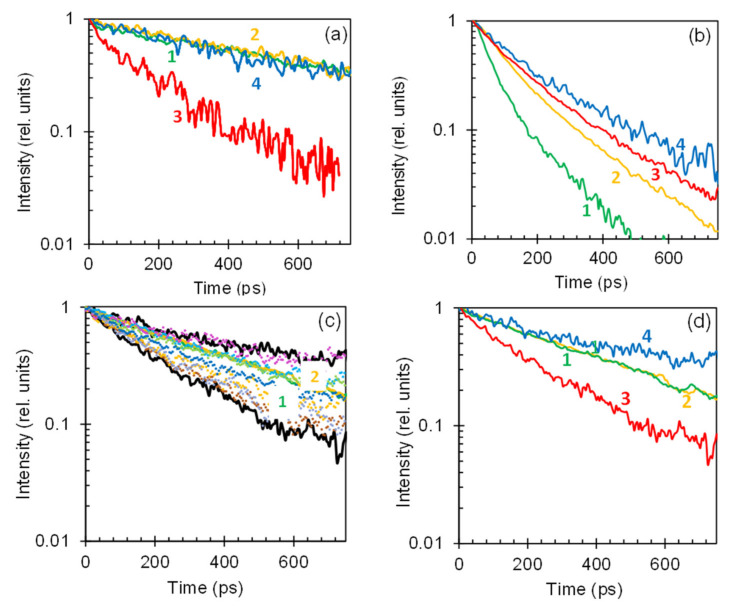
(**a**) Emission kinetics of HITC:PMMA at the low dye concentration equal to *n* = 3.2 g/L measured on top of glass (green trace 1), smooth thermally deposited Au film (yellow trace 2), and Au nanofoams: red trace 3 corresponds to the lower boundary of the emission kinetics spread, while the blue trace 4 corresponds to the upper boundary of the emission kinetics spread. (**b**) Same for the high dye concentration *n* = 36.5 g/L. (**c**) Emission kinetics of HITC:PMMA (*n* = 12.7 g/L) recorded in different local spots of the dye-doped Au nanofoam samples. Traces 1 and 2 are the emission kinetics of the *n* = 12.7 g/L HITC dye on top of glass and smooth Au substrates, respectively. (**d**) Same as 3a and 3b at the dye concentration *n* = 12.7 g/L.

**Figure 4 nanomaterials-10-02135-f004:**
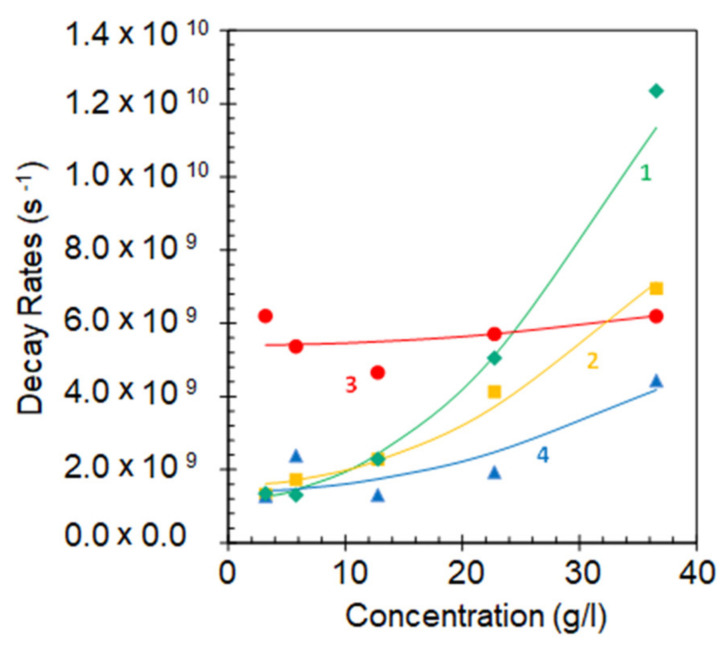
Emission decay rates in HITC:PMMA films at different dye concentrations on top of glass (trace 1), smooth thermally deposited Au film (trace 2), sub-volumes of the Au foams characterized by short emission kinetics (trace 3), and sub-volumes of the Au foams characterized by long emission kinetics (trace 4). Solid lines are the fits with the formula (*A* + *W*) + *γn*^2^. The spread of emission kinetics is a random process. This may explain larger error bars and poorer fit of the data points forming trace 3.

**Figure 5 nanomaterials-10-02135-f005:**
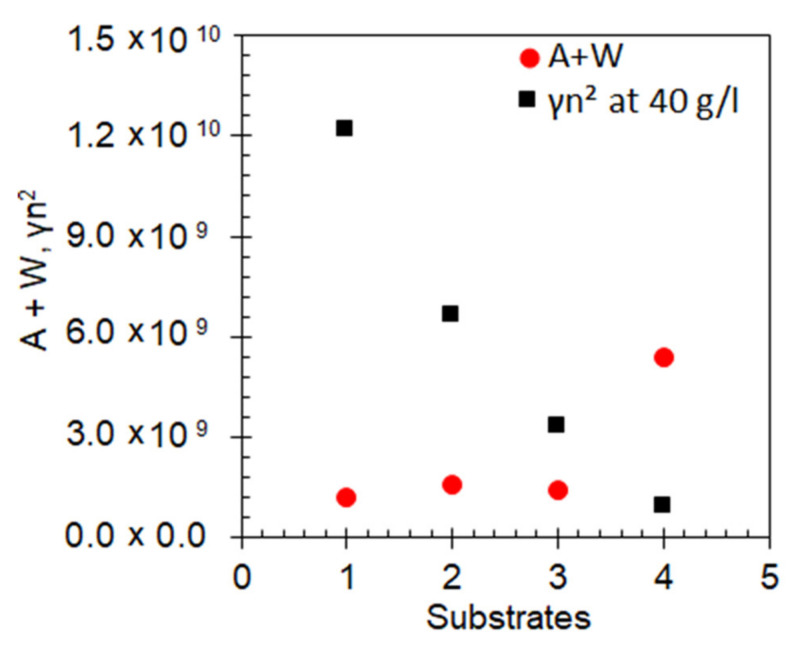
The values *A + W* (red circles) and *γn*^2^ (at *n* = 40 g/L, black squares) obtained from fitting of the datasets in Figure 4. (1) glass substrate, (2) smooth Au film substrates, (3) Au nanofoam (upper boundary of the emission kinetics spread), (4) Au nanofoam (lower boundary of the emission kinetics spread). The error bars are comparable to the sizes of the characters.

**Figure 6 nanomaterials-10-02135-f006:**
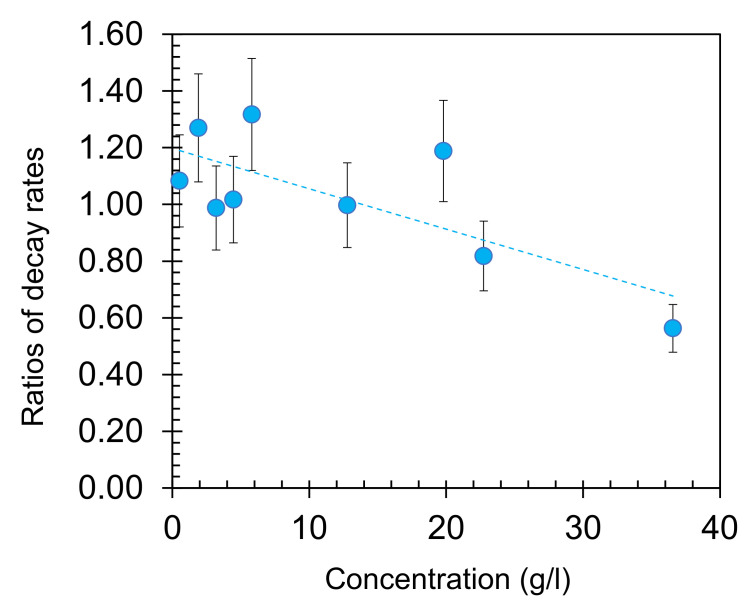
The ratios of the emission decay rates measured on top of smooth thermally deposited Au films and on top of glass, plotted as the function of the dye concentration. The dashed line is the linear fit of the data points—guide for eye.

**Figure 7 nanomaterials-10-02135-f007:**
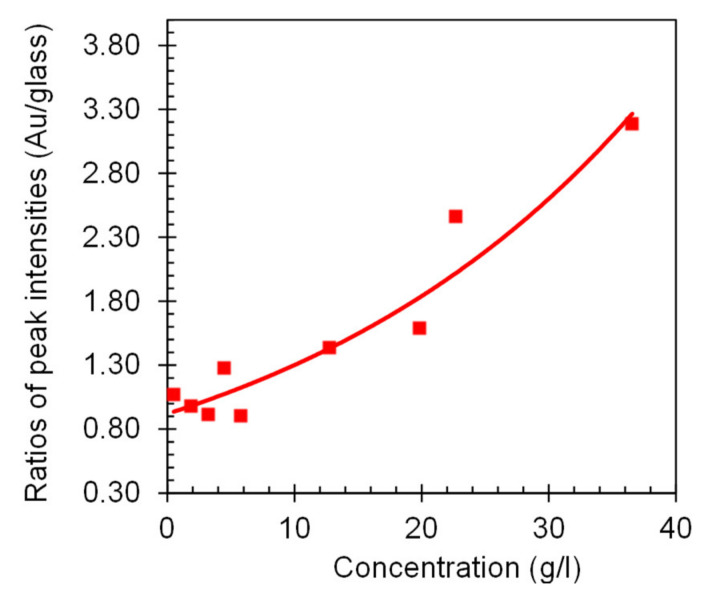
Ratios of the maximal emission intensities in HITC:PMMA films deposited on smooth Au films and on glass. The solid line is the fit of the data points with the second order polynomial—guide for eye.

**Figure 8 nanomaterials-10-02135-f008:**
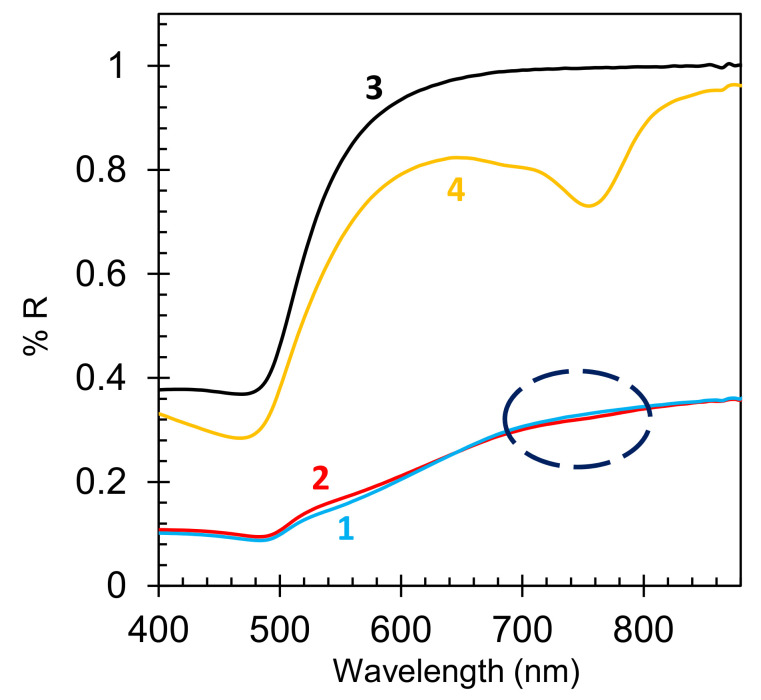
Reflectance spectra of the Au nanofoam without dye (trace 1) and with HITC:PMMA (*n* = 36.5 g/L) (trace 2). Reflection spectra of smooth gold film without dye (trace 3) and with HITC:PMMA (*n* = 36.5 g/L) (trace 4). The ellipse shows the area, where the dip in the reflection spectrum of Au nanofoam, caused by the absorption of the HITC dye, is expected.

**Figure 9 nanomaterials-10-02135-f009:**
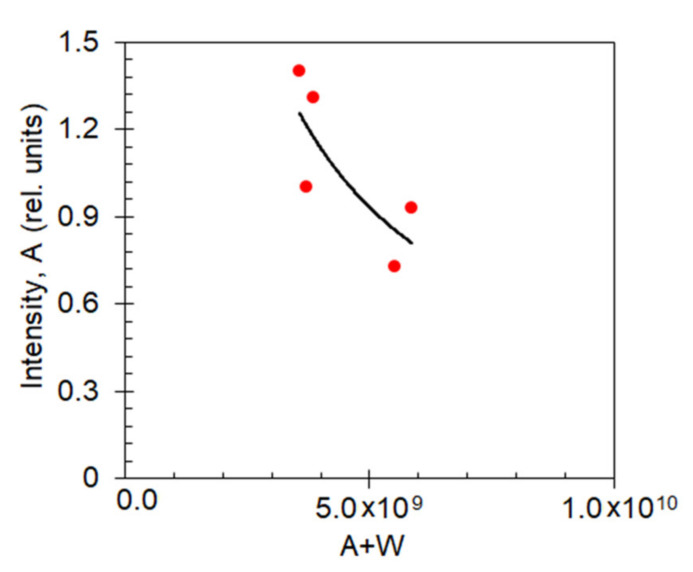
Maximal emission intensities (~*A*) plotted against the emission decay rates *(A + W)* measured in multiple local spots of the Au nanofoam sample at HITC concentration equal to *n* = 3.2 g/L. The solid line is the fit of the data points with the second order polynomial—guide for eye.

**Figure 10 nanomaterials-10-02135-f010:**
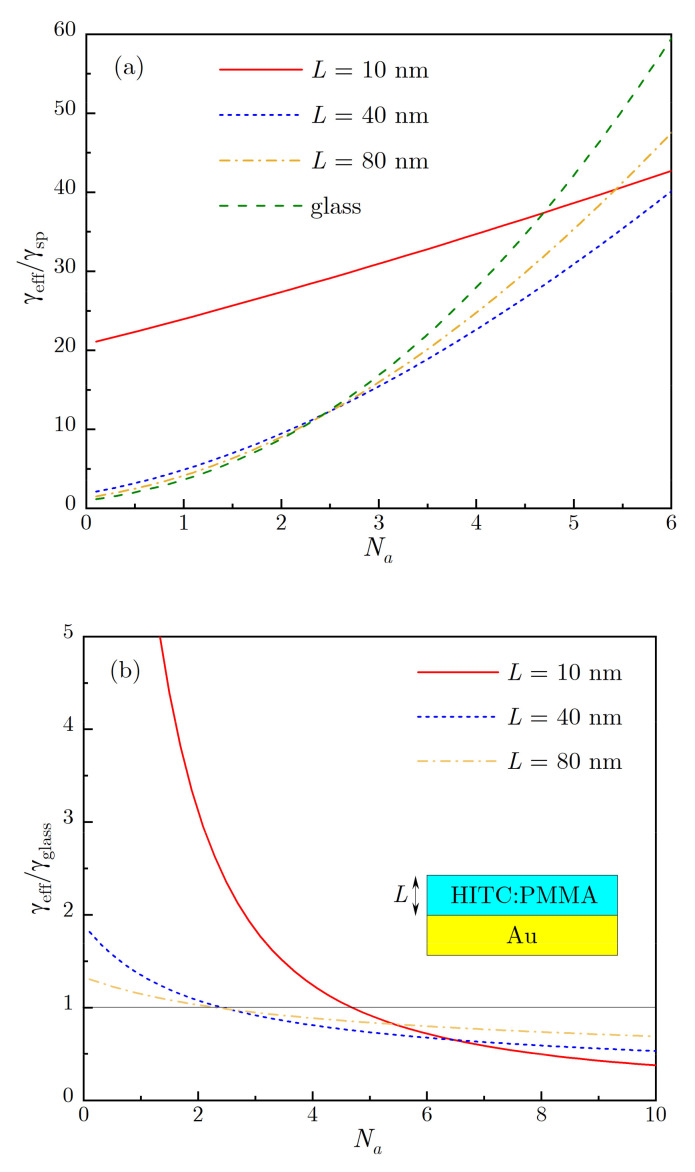
(**a**) Calculated effective decay rate, normalized by free-space spontaneous decay rate, plotted against acceptor concentration for several HITC:PMMA film thicknesses. (**b**) The ratio of effective decay rates for HITC:PMMA film on top of metal and of glass substrates is plotted vs. concentration for several film thicknesses.

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
