# Peer review of "Effect of Random Nanostructured Metallic Environments on Spontaneous Emission of HITC Dye"

_nanomaterials, 2020, doi:10.3390/nano10112135_

Round 1

Reviewer 1 Report

In this manuscript, the authors describe the results of their experimental study aimed at understanding the effect of metal surface on the emission kinetics of HITC dye dispersed in the PMMA matrix, deposited either on smooth or on porous Au films. They demonstrate the inhibition of the concentration quenching near metal accompanied by the higher relative emission peak intensity on gold compared to glass substrates at high dye concentration. The observed phenomena are explained by the reduction of the Forster radius in vicinity of metal, leading to the reduction of the concentration quenching and the enhancement of the energy transfer to metal.

The paper is nicely written, with providing enough experimental details and theoretical considerations. However, there are a few questions that need to be clarified before accepting this manuscript for publication.

  1. In Figure 7 the up to 3.3 times higher initial emission intensity of HITC:PMMA is demonstrated on smooth Au compared to glass at high concentrations of the dye. This fact is qualitatively explained by the reduction of the emission decay rate. However, the results could also be explained by the increased excitation rate near highly reflective Au surface. The authors are encouraged to elaborate on that.
  2. The emission spectra are not shown. Is there any change due to the presence of gold?

Additionally, there are several typos and

  1. Line 86: should be "γ" instead of "g"
  2. Figure 5: Title of Y axis should be “A+W, γn2
  3. Line 138: “excited” instead of “exited”
  4. Line 160: Au instead of Ag
  5. Line 227: what is “three-emission kinetics”?

Reviewer 2 Report

In this experimental paper, Zhu and co-workers study the emission kinetics of HITC laser dye deposited on different substrates. The experimental results (on luminescence, emission intensities, reflectance spectra) are described theoretically accounting via Förster energy transfer theory.

While the topic of the paper is suitable for the journal, currently the synthesis between the theory and experimental data is lacking. For example, it is difficult to track whether the theoretical curves are really theory or are just fits.

I recommend the authors revise their manuscript to improve the link to theory.

1) Why does the data in Fig 2 not cover the full range of 600-900nm?

2) In Fig. 4, the fitting of the “data 3” to the red line does not seem to be reasonable. Can the reason for the disagreement be explicitly addressed in the manuscript?

3) Where do the trend lines come from in Fig.6 and Fig. 7 and Fig. 9? Is it theory? Is it fitting? What is the fitting function and how many free parameters are there?

4) What is the meaning of the dashed line in Fig. 8?
